# Estimating spatiotemporally varying malaria reproduction numbers in a near elimination setting

Isobel Routledge [1], José Eduardo Romero Chevéz[2], Zulma M. Cucunubá[1], Manuel Gomez Rodriguez[3], Caterina Guinovart[4], Kyle B. Gustafson[5], Kammerle Schneider[4], Patrick G.T. Walker[1], Azra C. Ghani[1] & Samir Bhatt[1]

In 2016 the World Health Organization identified 21 countries that could eliminate malaria by 2020. Monitoring progress towards this goal requires tracking ongoing transmission. Here we develop methods that estimate individual reproduction numbers and their variation through time and space. Individual reproduction numbers, $R_c$, describe the state of transmission at a point in time and differ from mean reproduction numbers, which are averages of the number of people infected by a typical case. We assess elimination progress in El Salvador using data for confirmed cases of malaria from 2010 to 2016. Our results demonstrate that whilst the average number of secondary malaria cases was below one (0.61, 95% CI 0.55–0.65), individual reproduction numbers often exceeded one. We estimate a decline in $R_c$ between 2010 and 2016. However we also show that if importation is maintained at the same rate, the country may not achieve malaria elimination by 2020.

[1] MRC Centre for Global Infectious Disease Analysis, Department of Infectious Disease Epidemiology, Imperial College London, London W2 1PG, UK. [2] Ministry of Health (MINSAL), Calle Arce No.827, San Salvador, El Salvador. [3] Max Planck Institute for Software Systems, E1 5, Campus, 66123 Saarbrücken, Germany. [4] MACEPA, PATH, Seattle, Washington 98121, USA. [5] Institute for Disease Modeling, Bellevue, WA 98005, USA. Correspondence and requests for materials should be addressed to I.R. (email: i.routledge15@imperial.ac.uk)

Great strides have been made since 2000 in reducing the burden and mortality of malaria. The World Health Organisation (WHO) estimates that 57 out of the 106 countries with endemic malaria transmission in 2000 reduced their incidence of malaria by >75% between 2000 and 2015[1]. As a result, malaria elimination at the national level, defined as the absence of local transmission within a country[2], is now one of the targets in the WHO Global Strategy for Malaria 2016–2030[3]. In 2016 the WHO identified 21 countries for which it would be realistic to eliminate malaria within the next five years[4].

As countries attempt to move towards malaria elimination, tracking progress through quantifying changes in transmission over space and time is key. This information is necessary to effectively target resources to remaining 'hotspots' and 'hotpops'[5] where transmission remains, decide if and when it is appropriate to scale back interventions, and to evaluate the success of existing interventions. However, as countries approach zero cases, increasing focality in transmission and the impact of imported cases pose challenges to both reaching elimination[6] and measuring progress towards that goal. Increased spatial and temporal heterogeneity in malaria cases[7–9] in low transmission settings reduces the usefulness of national or regional level trends in incidence or prevalence, which can mask small areas of high transmission intensity. Furthermore, end-game surveillance and control measures are increasingly expensive per case. Therefore, while interventions must be targeted efficiently to be cost-effective[7,8,10], the identity of areas driving remaining transmission and their stability over time are poorly understood.

A wide variety of contextually varying factors have been hypothesised to drive transmission in low transmission settings, including increased risk in concentrated populations due to factors such as occupation (e.g., agricultural workers)[6], asymptomatic individuals acting as reservoirs of infection[11,12], changes in vector behaviour[13] and resistance to antimalarial[14] and insecticidal interventions[15]. Importation of malaria cases from neighbouring countries poses an additional challenge in many elimination settings. If many cases of malaria are imported, control measures may appear less effective due to small numbers of locally acquired cases arising from imported cases[16,17]. If there is sufficient importation, local cases can continue to occur even when the reproduction number of malaria under control, $R_c$, is below 1. Conversely areas with a high underlying $R_c$ but little importation may see sudden outbreaks of cases following a rare importation event due to their receptivity to malaria infection[18]. Challenges arise in measuring the sustainability of elimination[6,17], both in terms of quantifying the impact of control measures on transmission in the lead up to elimination, and in determining the risk of resurgence once elimination is achieved[19–21]. This information is also important when deciding if, when, and how to scale back intervention and surveillance methods[19].

Meeting these challenges requires measuring changes in transmission, often at fine spatial scales. However, existing methods used to quantify malaria transmission are poorly suited to elimination settings[9]. Parasite prevalence rates (PR) are not accurate below a PR of 1–5%[22,23] due to the large sample sizes necessary for precise estimates at low prevalence. The entomological inoculation rate (EIR), often seen as the 'gold standard' in measures of transmission intensity, is not reliable when transmission is highly focal and potentially unstable since EIR is very sensitive to heterogeneities in vector populations[24,25]. Use of serological data, while promising[26–28], is not currently feasible for use in many near-elimination contexts, as suitable cross-sectional survey data and/or appropriate markers to determine changes in malaria transmission are not available in all contexts.

A possible alternative, or complementary, measure of malaria transmission is the incidence of malaria cases, obtained through routine surveillance by Ministries of Health. Surveillance data are widely collected and sensitive to short term changes in transmission. While utilising these data can pose challenges, particularly in low-resource settings due to limitations in surveillance infrastructure and difficulty in establishing completeness of reporting, they can provide a wealth of information when such challenges are overcome. Individual level incidence data can be used to reconstruct the most likely pathways of transmission and estimate individual reproduction numbers, providing fine-scale insights into spatiotemporal transmission characteristics. While individual level surveillance data is often used in outbreak analysis of epidemic infections[29,30], robust methods are rarely applied to vector-borne diseases such as malaria, with a few notable exceptions[17,31,32].

Here we aim to estimate individual reproduction numbers over time and space by adapting methods from the study of information diffusion processes. These methods address the general problem of reconstructing information transmission using known or inferred times of infection by a 'contagion'[33–36]. They provide an adaptable framework to integrate multiple data types[37], identify likely unobserved cases/external infection sources, and have been evaluated using real and simulated transmission processes at multiple scales and network structures[36].

El Salvador provides a promising context to explore this approach. In 1980, El Salvador had the highest incidence of malaria amongst all Mesoamerican countries—with 95,835 cases and a 38% share of all cases in Mesoamerica. However, by 1995, the country contributed just 2%, maintaining low incidence until the present day. The country is now in the elimination phase and saw seven malaria cases in 2015 (0.1% of cases in Mesoamerica)[38]. Epidemiologists in El Salvador have kept records at a high spatial and temporal resolution throughout their malaria control and elimination efforts. In addition there has been a long history of reactive and active case detection, testing and treating all patients with fever with antimalarials and an extensive network of community malaria workers has been in place since the 1950s[38], evidence suggesting that case detection and treatment is strong. A full understanding of elimination in El Salvador could therefore provide useful insights for other countries as they aim to achieve and sustain elimination. Using the epidemiological line-list maintained by the Ministry of Health, we applied our methods to these data to estimate how transmission varied over space and time in El Salvador between 2010 and 2016.

Our results suggest a decline in $R_c$ between 2010 and 2016, with seasonal peaks during the wet season and during holiday periods. However we find that, based on the observed distribution of $R_c$ over time, with individual reproduction numbers often exceeding one $R_c > 1$), we cannot say with 95% confidence the country will achieve malaria elimination by 2020, assuming that importation is maintained at the same rate. Our results illustrate the role of importation in driving transmission dynamics in this country and provide independent estimates of the likelihood that El Salvador can eliminate malaria by 2020.

## Results

**Network reconstruction and estimated $R_c$ values.** Between 2010 and the first two months of 2016, a total of 91 cases of malaria were confirmed by microscopy in El Salvador, of which 30 were classified as imported. There were a total of six cases of *P. falciparum*, all of which were imported. Our estimated transmission network is shown in Fig. 1. Overall, the temporal dimension, informed by the prior distribution for the serial interval (Fig. 1a), dominates the identification of infector-infectee pairs (Fig. 1b). We identified two locally acquired cases which could not be plausibly linked to other cases within the dataset (Fig. 1c). These

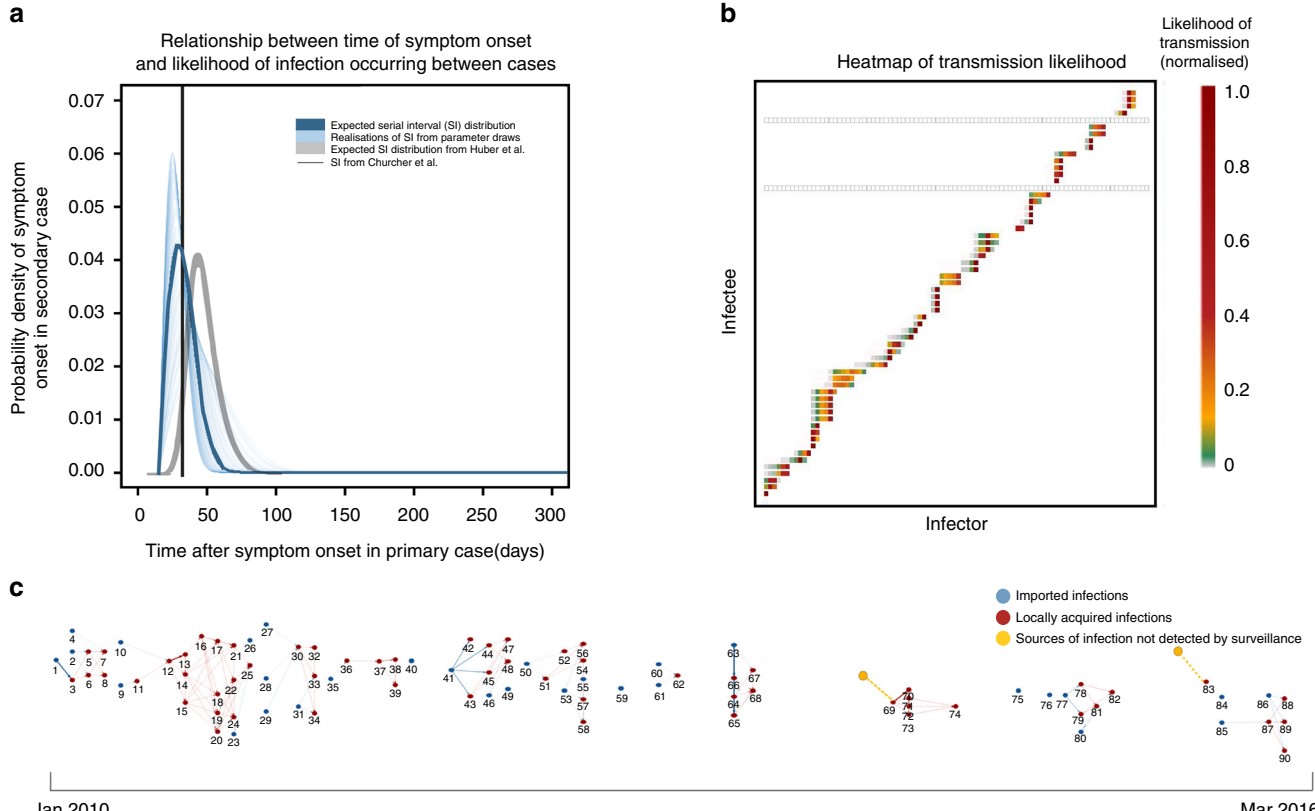

**Fig. 1 a** Serial interval (SI) distribution used in the analysis. Thin blue lines represent 300 realisations of the SI distribution resulting from draws from the distributions of the parameters determining the shape of the SI and incubation period. The serial interval distribution is the distribution of the time between the onset of symptoms (fever) in a case and the onset of symptoms in the case(s) it infects. The thicker blue line represents the expected SI distribution. For comparison, the grey line represents the SI distribution estimated for symptomatic, treated *P. falciparum* infection from[47] and the black line shows the expected SI for *P. falciparum* from[17]. **b** Heatmap showing likelihood of transmission occurring between infector and infectee pairs. The *X* axis represents all possible infectors (all reported cases) of the observed cases, organised by symptom onset date. The *Y* axis represents all possible infectees (all locally acquired cases, as by definition we assume imported cases were infected outside of the country). Each square represents a potential infector/infectee pair. The colours of the heatmap represent the normalised likelihood of infector j having been the infector of infectee i. where red is 1 and grey is 0. Grey squares show where cases were not likely to be infected by to any observed case, and therefore presumably infected by an individual who was not detected by surveillance. These could be asymptomatic or unreported clinical cases. **c** Reconstructed network, where numbers represent the ID of cases in temporal order. The strength of likelihood of connection represented by weight of edges linking cases. The two locally acquired cases identified to be infected by unobserved sources of infection are highlighted

were estimated in periods in which a clear gap in the data was apparent, and may therefore represent unidentified importations, relapse cases or unreported locally acquired sources of infection.

We estimated the mean individual reproduction number over 2010–2016 to be 0.61 (95% CI = 0.56,0.65). This is consistent with the ratio of locally acquired to total cases (61:91 = 0.66), which has been proposed elsewhere as an approximate estimate of $R_c$[2]. When fitting a generalised additive model (GAM) to the data, the overall trend was a decline from a fitted $R_c$ of 0.73 at the start of the observation to 0.47 by the end of the period (Fig. 2). Individual reproduction numbers showed seasonal fluctuations through time, with regular peaks observed in December, which coincides with the end of harvest season for many crops in El Salvador and Guatemala, and August, which coincides with a period of national holiday and the end of the rainy season.

**Spatial distribution of cases and $R_c$.** Data were highly focal, with 70% of cases originating from two adjacent administrative departments neighbouring Guatemala, and 32% of cases originating from just two municipalities within these regions (Jujutla and Acajutla) (Fig. 3a, b). This pattern was also reflected in the

spatial distribution of $R_c$. While most areas of the country are predicted to have a low risk of $R_c$ reaching above one over the time observed, several regions have a much higher predicted risk of $R_c > 1$ (Fig. 3c). In these regions, the majority of cases imported into the region could be expected to result in at least one onward transmission event. However it is important to note that uncertainty in these predictions is high in areas where we have not seen cases. The area where we have the least uncertainty in our estimate, around the borders of Guatemala, suggest that most cases occurring there did not contribute to onward transmission.

**Impact of imported cases on transmission.** The mean marginal gain to the likelihood of including infections from imported cases into the constructed transmission networks was much higher than including locally acquired cases (0.081 compared to $3.44e^{-7}$), suggesting that imported cases are a major driver of transmission. Visual inspection of the most likely chains of transmission (Fig. 1c) also are suggestive of this, where the index case in a cluster of linked cases was almost always an imported case.

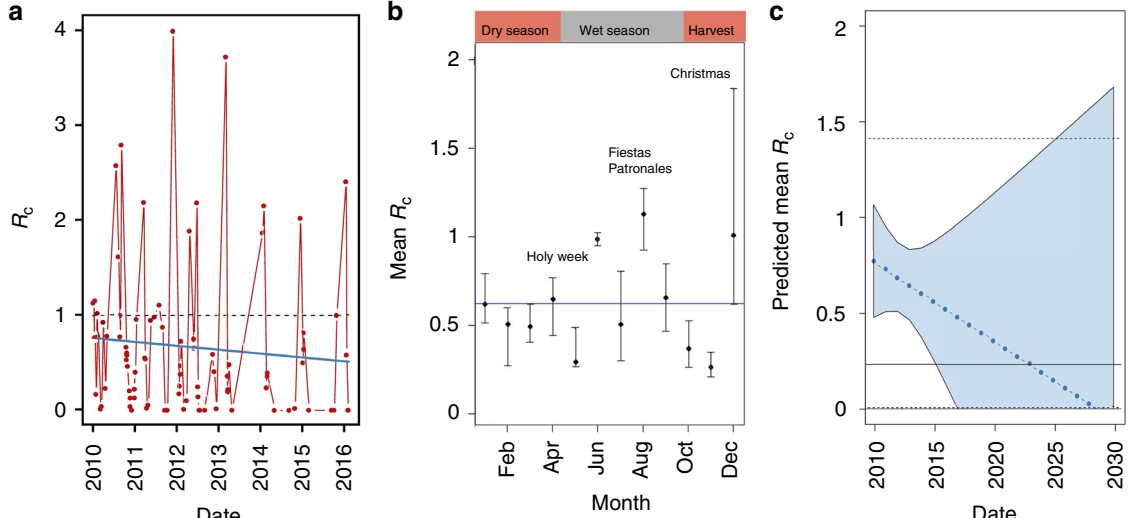

**Fig. 2 a** Individual reproduction numbers plotted over time. Individual, or case reproduction numbers ($R_c$) are the estimated number of individuals a given case is likely to have gone on to infect. Dashed line shows $R_c = 1$, blue line shows fitted Generalised Additive Model. **b** Posterior estimates of $R_c$ by month of year. Bars show 95% credible interval. Blue line shows the mean estimated $R_c$, the individual reproduction number, for the observation period. Key holidays, seasons and agricultural patterns are labelled. **c** Extended trendline to 2030 showing predicted $R_c$ Shaded area shows 95% credible interval from prediction and solid line shows mean threshold of $P = 0.05$ of cases occurring with an $R_c$ above one. Dashed lines show 97.5 and 2.5th quartiles for this threshold from 10000 simulations

**Endgame predictions based on $R_c$ and stochasticity**. To investigate potential timelines to elimination (i.e., the absence of local transmission) we characterised heterogeneity in the reproduction number using a Gamma distribution which, when fitted to the data, suggests a threshold mean $R_c$ of 0.22, below which there would a <5% chance of any individual reproduction number exceeding one. Using our fitted trend in the mean $R_c$, we expect this level to be reached by 2023, assuming no change in the rate of importation (Fig. 2c).

## Discussion

Understanding how transmission varies over time and space is critical to efforts to achieve and maintain elimination of infectious diseases such as malaria. Reconstructing transmission chains and estimating individual reproduction numbers has been used widely within epidemiological analysis[30,39,40], but rarely used to study vector-borne or endemic diseases, albeit with a few notable exceptions[31,32]. Separately, similar problems have been approached within human social network analysis, through a family of approaches known as independent cascade models[33–36]. Here we have adapted these methods to routine data from an eliminating Central American context, El Salvador, in order to inform progress towards their malaria elimination goals.

Our results suggest that the time-averaged $R_c$ has been below 1 in El Salvador since 2010, suggesting that sustained endemic transmission at the country level has already been interrupted. However, we estimated individual reproduction numbers exceeding one, resulting in ongoing outbreaks of transmission. Assuming the downward trend observed in $R_c$ between 2010 and 2016 continues, we expect the probability of such outbreaks to be <5% by 2023 if current levels of malaria importation and control continue. However, because we found imported cases to have higher reproduction numbers and their inclusion in the transmission tree increased the overall likelihood of the tree much more than locally acquired cases, it is important to note that the rate of importation is likely to affect the distribution of $R_c$. With increased importation this timeline to elimination could lengthen. Conversely, if importation was reduced, the timeline would be

shortened. Thus the levels of malaria importation from neighbouring countries would likely need to be decreased in order to achieve elimination by 2020, following current WHO certification policy of three years of zero locally acquired cases.

Given the extensive surveillance of migrants already carried out by El Salvador, as well as the free-movement and trade agreements which exist between El Salvador, Guatemala, Honduras and Nicaragua, the most efficient way of achieving this is likely to be through reducing the prevalence of malaria throughout Central America. However, given the seasonal peaks in $R_c$ we estimated to occur in August and December, which coincide with national holidays and the end of harvest season, there could additionally be an opportunity to increase surveillance activities and interventions during these key periods of high human mobility.

The Elimination of Malaria in Mesoamerica and Hispaniola (EMMIE) initiative aims to eliminate local malaria transmission from the entire Mesoamerican region by 2020[41]. Our results support the need for a regional approach to elimination. The clear impact of importation in driving residual transmission in El Salvador highlights the need for cross-border collaboration. In order to drive transmission down, areas of the highest 'receptivity' to intervention and 'vulnerability' to importation of cases must be identified. Approaches such as ours, which map transmission risk, could be combined with information about human movement to identify foci for increased surveillance, vector control and other interventions. Our approach using El Salvador as a case study could be adapted and used in other Central American countries or other contexts aiming for elimination.

We identified two cases with no clear source. When raising the threshold likelihood for linking observed cases as part of our sensitivity analysis and reducing the number of possible edges in the network, we find 7 missing cases. There is evidence in some low transmission contexts, especially where rapid declines of malaria have been seen recently, of significant asymptomatic and/or submicroscopic reservoirs of infection which may transmit to onwards transmission[42]. These could be sources of the missing infections identified in our study. However, El Salvador is unlikely to have a large amount of asymptomatic cases due to a long

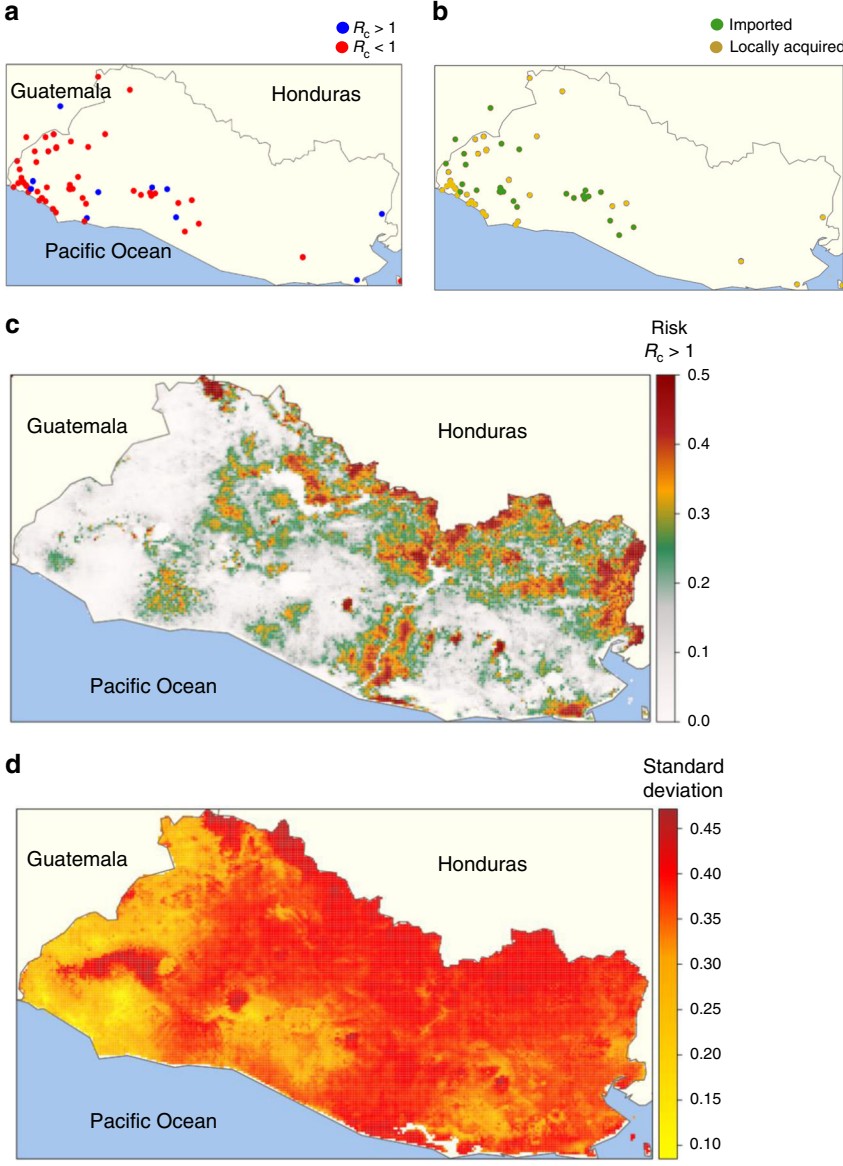

**Fig. 3 a** Distribution of $R_c$ values by location of residential address. Red points represent an $R_c$ (individual reproduction number) below one, blue points represent an $R_c$ value above 1. **b** Distribution of imported and locally acquired cases by location of residential address. Yellow points represent locally acquired cases, green points represent imported cases. **c** Map of risk of $R_c$ exceeding 1 if a case were to occur in an area. Note this estimate does not consider risk of importation, but estimates receptivity to transmission if importation were to occur. **d** Standard deviation in risk estimates from **c**

history of low numbers of cases. If our missing source of infections was mainly indigenous asymptomatic infections, it would signify that there is an asymptomatic reservoir contributing to onward transmission and that must be controlled to reach elimination. This could be achieved through PCR-based screening and treatment or increased vector control in focal areas. An alternative explanation is that there may be a small number of unreported symptomatic cases or relapse cases which were not reported or detected, which could be indigenous or imported. If due to importation this would further support the need for strong regional cooperation via initiatives such as EMMIE to reduce burden in neighbouring countries, and to maintain vigilance over extended periods in a near-elimination stage.

There are several limitations to this work. First, while we use epsilon edges to identify potential external sources of infection, this approach is only appropriate for smaller numbers of missing cases. Given the long history of small numbers of cases and testing and treating ~100,000 febrile patients per year (of which

only 6 were positive for malaria in 2015), and the programme of active case detection we feel this is a reasonable assumption, however in other contexts this may be a larger concern and methods such as reversible jump MCMC methods for data augmentation and inference may be appropriate.

Second, by the nature of a near elimination context our sample size is very small. The methods we use for estimating $R_c$ are well suited to small, well observed infection cascades, however this small sample size does provide a limitation for mapping, meaning our maps have relatively high levels of uncertainty outside of the areas of El Salvador where cases are seen principally around the pacific coast, Guatemalan border and in San Salvador. There is scope to incorporate expert knowledge to refine the map in areas where data are lacking.

It is important to consider whether methods presented here can be used in low resource settings that are earlier in the elimination process. In these contexts the number of cases is likely to be higher and there may be less complete reporting data and

potentially a higher reservoir of asymptomatic infection. In order to address these challenges several adaptations to the methods presented here may be required. First, there may be a need to incorporate more sources of information, e.g., demographic, spatial and possibly genetic data[30,37]. Second, Bayesian data augmentation techniques[43] may be required to explore the implications of large amounts of missing infection and potential reporting biases. In the case of more asymptomatic or untreated malaria there may be more uncertainty in the serial interval of malaria, however using our current approach can propagate this uncertainty through the model. Generalisations to full likelihood based or Bayesian hierarchical inference[36] can be beneficial by providing flexibility through parametric forms by allowing for the incorporation of additional factors (e.g., genetic distance) specific to the disease and context.

This work provides a novel framework for making use of routine surveillance data, and allows quantification of malaria transmission and its variation over space and time in contexts where traditional methods such as parasite prevalence are unsuitable. This is key in designing optimal strategies to accelerate, achieve and maintain elimination. To apply to other contexts several adaptations and extensions may be required. Firstly, in this dataset there were no confirmed relapse cases, however in many contexts we may see *P. vivax* relapse, in which case the algorithm could be adapted to allow for a likelihood for 'reinfection' or a looped network edge. Second, in settings where transmission links are less clearly identifiable or different data sources are available, this approach can be adapted to incorporate additional features such as spatial or genetic distance weightings into the likelihood function[37], following on from work based on Wallinga and Teunis approaches[30,43,44]. Finally, asymptomatic reservoirs and causes of missing cases, as well as their impact on transmission dynamics could be explored in more detail to consider surveillance system design and evaluation of its strength.

In conclusion, this work adapts concepts from network theory to build and apply novel methods to map transmission over space and time in a near-elimination setting, using only routine malaria surveillance data. Such approaches offer opportunities to better understand transmission dynamics and their heterogeneities in near elimination settings to better target interventions for elimination. We estimated timescales for reaching elimination and clarified the effect of importation on the speed and feasibility of achieving and maintaining zero cases. In the context of El Salvador, our results highlight the impact of importation on sustained transmission and highlight the need for cross-border collaboration. Our approach could be useful in a wide range of contexts where good quality routine surveillance data are collected, such as outbreaks and endemic diseases nearing elimination.

## Methods

**Data**. The data, obtained from the Salvadorian Ministry of Health (MINSAL), consisted of all confirmed cases of malaria between 2010 and the first two months of 2016 ($N = 91$ cases, of which 30 imported, 6 *P. falciparum*, 85 *P. vivax*). All but two cases had an address listed. For these cases the location was available at the *municipio*, or municipality level, and the coordinates of the centroid of the municipality (which for both were cities) were used as the geo-location. Two cases had addresses listed outside of El Salvador (in Guatemala). The latitude and longitude of cases with residential addresses in El Salvador ($N = 85$) were found to caserío (hamlet level) using Open Street Map (https://www.openstreetmap.org/) (Supplementary Note 1). Ethical approval was not required for analysing these data as the data were not individually identifiable and could not be traced back to a single person or household. This is because the patient's residence was geo-located to hamlet level, creating a buffer and meaning we do not identify an individual's exact residential location.

Data were captured through El Salvador's national epidemiological surveillance system (VIGEPES). These include cases reported by 30 public hospitals, 746 health facilities and thousands of community health workers stationed throughout the country (~3246 in 2010)[38,45]. During this period, the number of blood slides tested

per year remained similar (Supplementary Table 1). The line-list featured a unique patient identifier, address, age, sex, symptom onset date, and treatment seeking date, as well as details about treatment and diagnostic testing. All confirmed cases were treated. Detailed case investigation was carried out by MINSAL and cases were identified as imported or locally-acquired based on travel history, as well as primary, secondary, tertiary or orphan cases without clear sources, based on relationship with and proximity to previous cases. We obtained the latitude and longitude of the address, accurate to caserío (hamlet) level, using Open Street Map (https://www.openstreetmap.org/). El Salvador carries out reactive case detection following presentation at health facilities. However, in 2011, of 4500 slides examined through reactive case detection (representing 4.5% of all slides examined), just one additional case was detected. Both passive and active screening of migrants at key border crossings and in agricultural areas near borders also takes place. In these targeted areas, individuals are monitored for fever in the past 30 days, tested, and a single dose of chloro-primaquine prophylaxis is provided. In 2011, the Ministry of Health reported that 33,000 migrants were reached through active and passive case detection and an additional four cases of malaria were found[45]. Most cases were detected through passive surveillance in health facilities, at borders and by community health workers in rural areas.

**Serial interval distribution**. The serial interval is defined as the time between a given case showing symptoms and the subsequent cases they infect showing symptoms[46]. For a given individual $j$ at time $t_j$, showing symptoms before individual $i$ at time $t_i$, the serial interval distribution specifies the normalised likelihood or probability density of case $i$ infecting case $j$ based on the time between symptom onsets, $t_i – t_j$. The serial interval summarises a number of distributions including the distribution of (a) the times between symptom onset and infectiousness onset, (b) the time for humans to transmit malaria parasites to mosquito vectors, (c) the period of mosquito infectiousness, and (d) the human incubation period.

We defined a prior range of possible serial interval distributions for malaria. The serial interval distribution of treated, symptomatic *P. falciparum* malaria, previously characterised using empirical and model based evidence[47] was adapted to inform the prior distribution for the relationship between time and likelihood of transmission between cases in El Salvador. Two cases imported from West Africa were *P. falciparum*, however the remainder of cases were *P. vivax*. As a result the prior distribution was altered to better reflect the biology of *P. vivax* and the dominant vector species in El Salvador, *Anopheles albimanus*, but was uninformative enough to allow for possible variation in transmission dynamics, for example due to imported infections with *P. falciparum*. In addition, there is a possibility of a small number of asymptomatic or undetected and therefore untreated infections contributing to ongoing transmission, which will take on a longer serial interval. By defining a prior distribution for the serial interval distribution we can account for some of this uncertainty.

We use a shifted Rayleigh distribution to describe the serial interval of malaria, which can be varied by changing two parameters: $a$ and $\gamma$. The parameter $a$ governs the overall shape of the distribution, and $\gamma$ is the shifting parameter accounting for the incubation period between receiving an infectious bite and the onset of symptoms (Fig. 1a). The $\gamma$ shifting parameter was defined as ranging between 10 and 15 days to account for the minimum extrinsic incubation period within the mosquito and the minimum time between infection and suitable numbers of gametocytes in the blood to lead to symptom onset[48]. The prior for the $a$ parameter determining the shape of the distribution was given a Uniform distribution and bounded, giving an expected time between symptom onset of one case and symptom onset of the case it infects of 29 days (95%CI = 16–300 days, sd = ±7 days), with the lower bound having an expected serial interval of 25 days (95% CI = 16 – 299 days, sd = ±4 days) and the upper bound 47 days (95% CI = 16–300 days sd = ±18 days). By comparison the expected values for treated *P. falciparum* from existing literature range between 33[17] and 49.1 days (95%CI = 33–69)[47].

**Determining the transmission likelihood**. We assume cases were classified correctly from case investigation as imported or locally-acquired based on recent travel history. Following this assumption, locally acquired cases could have both infected others and been infected themselves. However imported cases could only infect others, as we assume their infection was acquired outside of the country. There were no confirmed relapse cases in the dataset, and all cases were treated with primaquine and chloroquine (radical cure) after being detected. Treatment is initiated before cases are confirmed by microscopy (see Supplementary Fig. 1). Active case detection is initiated locally following a confirmed case and in active foci in which local surveillance is believed to be weak. In these scenarios blood slides are examined within 24 h of being taken[49]. Given this, we assume that an individual can only be infected once by a case that has shown symptoms earlier in time.

Our data input consisted of a time series of symptom (fever) onset $t \in \{t_1, \ldots, t_n\}$, time ordered such that $t_1 < t_2, \ldots, < t_n$. While the times of symptom onset are known, the data do not indicate who infected whom and the underlying transmission chain, $\mathcal{T}$. The goal of our model is to infer the most probable network structure, $\mathcal{G}$, connecting these $n$ infections. We can view cases as nodes in a network $\mathcal{G}$, and possible transmission events as the edges linking nodes. We infer $\mathcal{G}$ solely from the symptom onset times $t$, a serial interval

distribution, and prior probability distributions for the serial interval distribution parameters.

$\mathcal{G}$ contains all possible spanning transmission chains over which an infection could spread given the observed times. $\mathcal{G}$ therefore includes the most likely transmission tree, but also includes, other possible trees supported by the data. We therefore can view a particular transmission tree $\mathcal{T}$ as a realisation of a stochastic diffusion process generated over an underlying network $\mathcal{G}$. Crucially, $\mathcal{G}$, accounts for competing edges and is sparse (only includes plausible edges).

For a given transmission tree $\mathcal{T}$ describing infection events linking cases and assuming the independent cascade model[33], the (upper triangular) likelihood of observing our times of symptom onset is simply the product of all permissible pairwise transmission likelihoods in the tree[35]. Our exposition until this point is the same as that introduced by Wallinga and Teunis[29] and extended to a wide variety of contexts by others[30–32,43,44,50]. However, in contrast to previous methods based on Wallinga and Teunis we maximise the likelihood $f(t|G)$ conditional on an underlying $\mathcal{G}$, a problem that is NP-hard[51]. Previous approaches have either allowed all possible connections in $\mathcal{G}$[29], sampled from the likelihood[52] or explored a limited number of pathways[53]. Here, by following the approach introduced by Rodriguez and Schölkopf[35], we find the most likely underlying transmission network given the timing of symptom onset for a set of $k$ transmission events linking cases. The computational hardness of maximising $f(t|G)$ meant that an optimal solution could only be found by exploring every possible transmission tree on $\mathcal{G}$. However, due to the submodularity of the independent cascade model[33] a near optimal solution could be found using a greedy algorithm. Briefly, the greedy algorithm used starts with an empty graph, and then add edges sequentially such that the marginal gain in the likelihood of the transmission tree for each iteration is maximised. The marginal gain measures of importance for each edge of the network through the gain that each edge provides to the total solution over competing edges, and therefore applies shrinkage to the raw pairwise likelihood with the likelihood of competing edges. We stop when we have reached $k$ edges (see Supplementary Fig. 2). Stopping at $k$ edges ensures that the resulting network is sparse which not only ensures a parsimony but removes unnecessary edges that could influence $R_c$ calculations. An appropriate value of $k$ is defined by adding edges until the marginal gain in likelihood of adding additional edges is below a given threshold (0.0005). We carried out a sensitivity analysis and find our results are robust to changes in this threshold between 0.001 and 1e−10 (Supplementary Note 2, Supplementary Fig. 3).

**Accounting for missing cases**. Assuming all cases reaching community health workers or health facilities are recorded, missing cases may be generated by two processes. Symptomatic cases may be missed by not seeking care or being found through active case detection, and or cases may be asymptomatic and therefore unlikely to seek care or be detected. The latter may have densities of parasites in their blood which are too low to be detectable by microscopy if active case detection occurs. These processes apply to both imported cases or locally acquired cases. We assume the pool of asymptomatic cases in the country is low and has a small contribution to ongoing transmission. To estimate the proportion of cases which may be going undetected within our independent cascade framework, we consider outside sources of infection, $\pi$ that represent unobserved individuals who can infect any observed individual, $i$, in a transmission chain. Every observed individual $i$ can get infected by unobserved individuals, $\pi$, with an arbitrarily small probability $\epsilon$. This so called $\epsilon$-edge is connected to every node in our network and do not, by design, participate in the diffusion propagation. The $\epsilon$-edges prevents breaks in the network diffusion cascade where the likelihood of transmission between observed cases is sufficiently low, and instead the case is linked to an external source. The specific value of $\epsilon$ was set at 0.0005, aiming to find a balance between false positives and false negatives when linking cases by infection events. The higher the value of $\epsilon$, the larger the number of nodes that are assumed to be infected by an external source.

**Estimating $R_c$**. We can establish individual reproduction numbers for each case by creating a matrix, $R$, where each column represents a potential infector and the rows represent a potential infectee, describing which infector edges are connected to infectees and the normalised marginal gain of that edge. Intuitively then, by taking the row sums of $R$ we get the (fractional) number of secondary infections and therefore a point estimate of the time varying reproduction number $R_c(t_j)$ This reflects for an individual, how many people they are likely to have gone onto infect. When multiple individuals have been infected at a given time and/or place, we can take the mean individual $R_c$ and uncertainty in this value as an indicator of reproduction numbers for a given time and/or location.

In contrast of traditional methods based on Wallinga and Teunis[29] using the marginal gain in this way encapsulates not only the pairwise likelihood of transmission between two cases, but conditions this likelihood on the impact of competing edges in the inferred network. Given the provable near optimal solution of the greedy algorithm and the use of marginal gains in calculating $R_c$, our estimates of $R_c$ provide more rigorous estimates of reproduction numbers than just using standard Wallinga and Teunis[29] approaches, which do not consider the overall transmission tree in optimisation and do not account for missing cases (see Supplementary Note 3 for full derivation of methods).

**Estimating timelines towards elimination**. To explore trends in $R_c$ over time, we fitted a generalised additive model (GAM) to the estimated $R_c(t)$ values and extended this line beyond the period of observation to 2030. We then also fitted Gamma, Power law and Exponential distributions to the estimated $R_c(t)$ values, and found they were best represented by Gamma distribution according to AIC scores. To explore the likelihood of elimination by a given time point, we randomly drew 10,000 $R_c$ values from Gamma distributions with increasingly small mean reproduction numbers, keeping the fitted shape parameter constant. We then found the threshold mean $R_c$ below which the probability of an individual $R_c$ exceeding one is <5%. By extending the current fitted trendline for $R_c$ values to 2030, we identified the expected timepoint for $R_c$ to reach this threshold value, given the observed decline in $R_c$ observed over the study period.

**Mapping $R_c$**. To map estimates of transmission risk, individual reproduction numbers were divided into those above and below one. The latitude and longitude of the reproduction numbers were included in a geospatial hurdle model implemented in rINLA[54] where demographic and environmental covariates were used to estimate the likelihood of a case having a reproduction number above 1 if imported into the area in 2010 (Supplementary Note 4, Supplementary Table 2)). This is a measure of malaria 'receptivity' or underlying transmission potential rather than overall malaria risk, as importation likelihood is not quantified in this analysis. AUC scores from leave one out cross validation were used to assess model fit (Supplementary Fig. 4).

**Code availability**. The source code used for this analysis are available from the authors upon reasonable request.

**Data availability**. Case data are not publicly available because they are nationally-owned data therefore the authors do not have the permission to host them but they are available from the authors upon reasonable request and with permission of the Ministry of Health, El Salvador (MINSAL).

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

## Acknowledgements

This work was funded by a studentship to IR from the Wellcome Trust. We additionally acknowledge Centre support from the UK Medical Research Council. We would like to thank Finlay Campbell for advice on network visualisation, Thibaut Jombart and Anne Cori for discussions surrounding transmission tree reconstruction and reproduction number estimation.

## Author contributions

Conceived study: I.R., S.B., A.C.G., P.G.T.W., M.G.R.; collected data: JERC (Ministry of Health, El Salvador); collated/obtained data: I.R., Z.C., C.G., K.S., P.G.T.W.; carried out analysis: I.R., S.B., K.G.; wrote paper: I.R., S.B.; commented on drafts/edits: I.R., S.B., A.C. G., P.G.T.W., M.G.R., Z.C., C.G., K.S., J.E.R.C., K.G.

## Additional information

**Competing interests:** The authors declare no competing interests.

