## [Peer Review File · Nature Communications]

Reviewers' comments:

Reviewer #1 (Remarks to the Author):

This is a well written paper that has relevance for those conducting malaria evaluation studies, developing surveillance strategies, and identifying relevant interventions for areas scaling up for elimination. The statistical analysis is appropriate; this work is reproducible, if one has high level statistical skills. To strengthen the paper, there are a couple of areas in need of further clarification. I have the following comments:

1. Line 39: the authors should briefly describe and cite the studies that have been done that describe factors driving or sustaining transmission in low transmission settings.
2. Line 44: this sentence is vague
3. Line 47: why is success is quotes?
4. Line 60: the authors should acknowledge and describe the challenges of using MOH data, especially in low-income countries.
5. Line 70: the meaning of this sentence is not at all clear.
6. Line 217: it would be useful to briefly describe how this method contributes to, or compliments, Bayesian methods.
7. A section describing the limitations of this method is needed.
8. The discussion section should also describe how low-income countries, with presumably less complete surveillance data could use this method.

In short, this paper would make a nice contribution to the malaria literature.

Joe Keating, PhD
Professor
Department of Tropical Medicine
Tulane University.

Reviewer #2 (Remarks to the Author):

Overall, this is an interesting paper tackling an important issue of how to measure and map transmission in very low transmission settings. I think it's a useful contribution to the literature. I have a few comments below.

Intro - I'm assuming this statement "parasite prevalence rates are not accurate below a PR of 1-5%" relates to the fact that large sample sizes are required to achieve precision in estimates at low prevalence? The authors might want to change their language to make that point.

Data

Could the authors provide any information on the completeness of the data? We addresses available for all cases? Were all addresses geocoded (lat/long) successfully?

Was any attempt made to correct for the fact that individuals living far from points of diagnosis (facilities, community health worker posts) are less likely to seek treatment? I realize the framework allows for missing case data, but when modeling this spatially the spatial bias in case data might affect the final map.

Mapping Rc

Why did the authors choose to treat this as a binary problem ($R_c > 1$ or < 1)? This aggregation seems to me to be losing valuable information, i.e. an R_c of 1.01 is not the same as an R_c of 5.

Were AUC values only generated for fitted predictions? Given the spatial prediction problem, I think it would be important to run some sort of (cross) validation.

Figure 1B - Its not clear to me what this is showing. Could the authors provide more explanation?

Figure 1C - I would move the legend to below the figure to avoid readers thinking the legend belongs to Figure 1B.

Figure 3 - as the authors make reference to Guatemala, it would be useful to show the location of El Salvador relative to its neighbors in Fig 3.

Many thanks for your insightful, constructive and helpful comments. We have responded to each comment and all changes to the text are highlighted as tracked changes in the revised manuscript and supplementary materials. Due to the word limit for the introduction, we have cut several phrases to allow for the inclusion of additions suggested by reviewers. The deleted sections can be seen in the tracked changes.

Reviewer 1

1. Line 39: the authors should briefly describe and cite the studies that have been done that describe factors driving or sustaining transmission in low transmission settings.

We thank the reviewer for noting this and have now added these references – see lines 40 to 43 of the revised manuscript.

2. Line 44: this sentence is vague

We have removed the line referring to vulnerability of malaria infection, as we agree it is vague. The phrase was initially included to introduce the term of “vulnerability” as used in WHO malaria elimination policy, defined as “Either proximity to a malarious area or frequent influx of infected individuals or groups and/or infective anophelines.”¹

3. Line 47: why is success is quotes?

We have now removed the quotation marks and changed success to sustainability (line 51). They were originally included to acknowledge that success has different definitions for different stakeholders – some view zero parasite prevalence as successful elimination, others define it as the absence of clinical cases².

4. Line 60: the authors should acknowledge and describe the challenges of using MOH data, especially in low-income countries.

These changes have been made, please see lines 68 to 71.

5. Line 70: the meaning of this sentence is not at all clear.

This reference was used to refer to the original contexts which this method was applied to, where a “contagion” may not be a disease but anything which can be propagated through networks (e.g. information diffusion along online social media, blog and mainstream media networks). Now changed (lines 79 to 80) as the reference to other forms of information propagation may be confusing.

6. Line 217: it would be useful to briefly describe how this method contributes to, or compliments, Bayesian methods.

Our approach naturally lends itself to Bayesian formulations. As it currently stands our formulation uses a proportional likelihood optimised by exploiting submodularity. However equivalent frameworks exist³ with explicit likelihoods derived from a survival view point of a temporal point process. These methods can be used within a full Bayesian hierarchical model. This is work that we are currently developing hope to release soon.

We have edited the section originally beginning on line 239 to incorporate a response to point 8 which clarifies how our method complements Bayesian methods (lines 247 to 250) , and also added a section to the supplementary information clarifying this with a similar paragraph to the one above (lines 143 to 146 of Supplementary Information).

7. A section describing the limitations of this method is needed.

A limitations paragraph has been added (lines 227 to 238).

8. The discussion section should also describe how low-income countries, with presumably less complete surveillance data could use this method.

We have also added this paragraph (lines 239 to 250). We discuss the challenges of possibly less complete reporting, larger numbers of cases and potential for a larger asymptomatic population; we introduce potential extensions to the methods (some of which are currently in development) including Bayesian approaches incorporating a wider range of data sources e.g. demographic and genetic data and data augmentation techniques to account for larger numbers of missing cases.

Reviewer 2

1. I'm assuming this statement "parasite prevalence rates are not accurate below a PR of 1-5%" relates to the fact that large sample sizes are required to achieve precision in estimates at low prevalence? The authors might want to change their language to make that point.

The reviewers' assumption is correct, the accuracy of prevalence rates relates to the samples sizes and precision. We agree that the statement "parasite prevalence rates are not accurate below a PR of 1-5%" could be made clearer and have now clarified this point (lines 57-58).

2. Could the authors provide any information on the completeness of the data? We addresses available for all cases? Were all addresses geocoded (lat/long) successfully?

The week but not day of symptom onset for three cases in 2010, so the midweek date (Thursday) was used in each instance. One case had a notification date but not symptom onset date provided so we drew many times from distribution of symptom to notification time for other cases to back calculate most likely symptom onset date.

Addresses were available for all but two data points (n=89). These points could be georeferenced to municipality, which in both cases was a city. Two cases had addresses in Guatemala.

All other cases (n=85) were geocoded to caserío (hamlet) level using a combination of open street map, google maps, bing maps and websites listing georeferences of caseríos/ lotificaciones /barrios (small collections of dwellings e.g. hamlet, housing development, neighbourhood) in El Salvador, organised by municipality. We cross checked locations and used fuller address information e.g. descriptors or street names when possible to avoid mis-identification. We have added a section to the supplementary information detailing the online resources used to georeferenced the residential addresses of cases (lines 34 to 45 of Supplementary Information, Note 2)

3. Was any attempt made to correct for the fact that individuals living far from points of diagnosis (facilities, community health worker posts) are less likely to seek treatment? I realize the framework allows for missing case data, but when modelling this spatially the spatial bias in case data might affect the final map.

We used a recently published accessibility index as part of a raster layer in our spatial model as a covariate (see Supplementary Table 2, line 197). We agree that is normally an important point and potential source of bias to consider, however the following features of El Salvador context make it less of an issue:

- Small size of country and extensive network of health workers in rural areas
- Active case detection in rural agricultural areas
- In addition, a small (n=152) PCR prevalence study found zero prevalence by PCR in school children in traditionally hyperendemic area of country, which suggests that in younger populations there is not a substantial reservoir of undetected infection.

Future work in other contexts could introduce a weighting e.g. using accessibility matrix and least cost paths to determine likelihood of a case being reported in a certain location

4. Why did the authors choose to treat this as a binary problem ($R_c >$ or <1)? This aggregation seems to me to be losing valuable information, i.e. an R_c of 1.01 is not the same as an R_c of 5.

We initially carried out the spatial analysis with a non-binary model, however we switched to a binary model for statistical reasons. Firstly the numbers of cases observed are so low, we have a zero inflation problem. Zero inflated likelihoods did not produce suitable models (as judged from WAIC and cross validation AUC).

More importantly, from an operational perspective, a key question for policy and programme design in near elimination settings is whether malaria will spread if introduced into an area, i.e. $R_c > 1$. Therefore the threshold of above and below 1 provides a clear, epidemiologically informed way of communicating risk. This approach has been used quite effectively by the authors and colleagues when communicating with Ministries of Health in other low transmission settings through the Clinton Health Access Initiative (CHAI).

5. Were AUC values only generated for fitted predictions? Given the spatial prediction problem, I think it would be important to run some sort of (cross) validation.

The AUC values reported are from leave-one-out cross validation – we have now clarified this in the main text (lines 425 to 426).

6. Figure 1B - Its not clear to me what this is showing. Could the authors provide more explanation?

We have now clarified and added more explanation, especially in defining x and y axes and the colour scale more clearly.

7. Figure 1C - I would move the legend to below the figure to avoid readers thinking the legend belongs to Figure 1B.

I am not sure if we have understood the reviewers comment fully but we think this is an artefact of how the pdf was made, as the image came after the legends. However “[insert figure one

approximately here]” is above the legends and would appear above the legends in the final manuscript. To clarify we have one legend for figure 1, referring to A, B, and C within the one legend.

8. Figure 3 - as the authors make reference to Guatemala, it would be useful to show the location of El Salvador relative to its neighbours in Fig 3.

We have edited figure 3 and also changed the figure legend in response to the changes.

References

1. World Health Organization. Global Malaria Programme. *Disease surveillance for malaria elimination : an operational manual*. (World Health Organization, 2012).
2. Cohen, J. M., Moonen, B., Snow, R. W. & Smith, D. L. How absolute is zero? An evaluation of historical and current definitions of malaria elimination. *Malar. J.* **9**, 213 (2010).
3. Alimi, T. O. *et al.* Prospects and recommendations for risk mapping to improve strategies for effective malaria vector control interventions in Latin America. *Malar. J.* **14**, 519 (2015).

REVIEWERS' COMMENTS:

Reviewer #1 (Remarks to the Author):

The authors have addressed the minor concerns I noted during the initial review process. I have no further comments. The paper has been improved.

Joe Keating

Reviewer #2 (Remarks to the Author):

The authors have satisfactorily responded to my comments. I have no further comments or suggestions.